# Development and Chemical-Sensory Characterization of Chickpeas-Based Beverages Fermented with Selected Starters

**DOI:** 10.3390/foods11223578

**Published:** 2022-11-10

**Authors:** Marina Mefleh, Michele Faccia, Giuseppe Natrella, Davide De Angelis, Antonella Pasqualone, Francesco Caponio, Carmine Summo

**Affiliations:** Department of Soil, Plant and Food Science (Di.S.S.P.A.), University of Bari Aldo Moro, Via Amendola 165/a, 70126 Bari, Italy

**Keywords:** alternative proteins, legumes, lactic acid bacteria strains, anti-nutritional factors, plant-based food, yogurt alternative

## Abstract

Legume protein ingredients are receiving continuous interest for their potential to formulate plant-based dairy analogs. In this study, a legume-based slurry was produced from an Apulian black chickpeas (BCP) protein concentrate and fermented with three starter cultures, *Streptococcus thermophilus* (ST), a co-culture of ST with *Lactococcus lactis* (STLL) and a co-culture of ST with *Lactobacillus plantarum* (STLP). The effect of fermentation on the biochemical, texture and sensorial parameters was evaluated. The same beverage without inoculum was used as a control (CTRL). All the obtained fermented beverages were characterized by high protein (120.00 g kg^−1^) and low-fat contents (17.12 g kg^−1^). Fermentation contributed to a decrease in the contents of phytic acid by 10 to 79% and saturated fatty acids by 30 to 43%, with the STLP fermentation exercising the major effect. The three culture starters influenced the texture and sensorial attributes and the profile of the volatile compounds differently. Fermentation increased the lightness, consistency, cohesivity and viscosity of the formulated beverages. On a sensorial level, STLL had a major effect on the acidity, sourness and astringency, while both ST and STLP affected the creaminess, solubility and stickiness. Legumes and grass aromas were masked in LAB-fermented samples, probably due to a new VOC formation. The functional properties of LAB fermentation, along with the high protein content of the black chickpeas concentrate, provide the opportunity to formulate a clean label and safe plant-based fermented beverage with higher nutritional value compared to the others currently found in the market.

## 1. Introduction

The global dairy alternatives market is expected to double its growth from 2021 to 2028 [1], and alternatives to milk (plant-based (PB) beverages) are the most popular dairy alternatives [2]. They can easily be distributed, and they allow the easy incorporation of nutrients, bioactive compounds and probiotics [3]. Today, plenty of PB beverages are available on the market, and the most used plants are almonds and soy. From a nutritional point of view, PB beverages made from nuts have a lower protein content (<1%) than conventional milk and yogurt (3.5%), while those made from legumes have a similar content [4,5]. The interest in foods that are high in protein is remarkably noticed, which triggered the food industry to start using protein isolates, such as gluten or pea isolates, as the main ingredients mixed with starch, oil and many other thickeners that could contribute to a good texture. However, the extraction of protein isolates is highly resource-demanding, and instead, researchers should better investigate the utilization of the green dry-fractionation technique. This latter allows for the recovery of protein concentrates with a protein content lower than that compared to a protein isolated (50–60 vs. > 80 g/100 g), but solely by physical methods, without using any water and/or chemicals [6]. 

However, dry-fractionated proteins are complex ingredients, and when obtained from pulses, their use is challenging due to the presence of anti-nutritional factors (ANFs) (e.g., phytic acid, tannins, saponins, α-galactosides, alkaloids, lectins and protease and chymotrypsin inhibitors) limiting the bio-availability of some minerals and decreasing the digestibility of proteins [7].

Many technological and biotechnological techniques were proposed to decrease the amount of ANFs and the off-flavors in food, and among them, lactic acid bacteria (LAB) fermentation was proved to be a successful method [8]. Moreover, LAB fermentation is recognized to naturally enhance the nutritional value, therapeutic benefits, sensorial properties and shelf-life of food [9]. To that end, bacteria of the genus *Lactobacillus*, *Lactococcus* and *Streptococcus thermophilus* are generally used [10,11], and their application in co-cultures seems to be advantageous and enhances the fermentation process [12]. In particular, the Lactobacillus and Streptococcus strains demonstrated masking or positively modifying the off-flavors of soybeans, lentils, lupins and pea protein isolates. Moreover, fermented foods are considered a functional food, which, today, are highly requested by people. 

The Apulian Black chickpea is a species historically cultivated in Apulia, a region in the south of Italy. It is physically similar to the species ‘Desi’ because of its black coat; however, on a genetic level, they are different [13]. This Apulian black type has a valuable nutritional composition, characterized by a high content of proteins, polyunsaturated fatty acids and antioxidants such as anthocyanins and carotenoids [14,15,16]. To limit the risk of genetic erosion of this valuable chickpea species, trials to formulate foods, such as pasta, bakery goods and puree containing Apulian Black chickpea flours, were successfully performed [15,17,18].

In this study, three techniques were used to optimize the characteristics of an Apulian black chickpea beverage; dry-fractionation of the chickpeas’ flour, heat treatment of the slurry at 90 °C and fermentation with selected LAB-culture starters. The protein concentrates of Apulian black chickpeas, hereafter called a BCP concentrate, were prepared and used as the main ingredient in the formulation of a clean-label fermented beverage. The slurry of the BCP concentrate was subjected to fermentation by selected LAB-culture starters isolated from plants. The influence of fermentation on the nutritional and VOC profiles and texture properties of formulated beverages was determined. A sensory evaluation was also performed. 

## 2. Material and Methods 

### 2.1. Materials

The protein concentrates from Apulian black chickpeas, with a protein content of 560 g kg^−1^ (d.m.), were obtained by dry-fractionation, provided by Innovaprot SRL (Gravina in Puglia, Italy), and produced from raw micronized flour of dehulled black chickpeas. Streptococcus thermophilus (LyofastV STV 10), Lactobacillus plantarum (Lactobacillus plantarum LPLDL^®^) and Lactococcus lactis ssp. Lactis (Lyofast VMO 01) were purchased from the company SACCO S.r.l (Cadorago, Italy). The three starters were certified to be vegan and isolated from plants. Sucrose was purchased from the market in Bari, Italy.

### 2.2. Products Formulation

Water was slowly added to the dry-fractionated BCP (5:1) while mixing and pre-heating slowly (Figure 1). The slurry was heated for 20 min at 75 °C. After cooling the mixture at 30 °C, sucrose (3.5% *w*/*v*) was added to the slurry and then heated again for 10 min at 90 °C [19]. The slurry was cooled down and then warmed up to 30 °C and divided into four aliquots; 

(1)CTRL, control sample not inoculated and stored at 4 °C.(2)ST, sample inoculated with 0.0025% (*w*/*v*) *Streptococcus thermophilus.*(3)STLL, sample inoculated with 0.00125% (*w*/*v*) *Streptococcus thermophilus* and 0.00125% (*w*/*v*) *Lactococcus lactis.*(4)STLP, sample inoculated with 0.00125% *w*/*v Streptococcus thermophilus* and 0.00125% *w*/*v Lactobacillus plantarum*.

According to the LAB fact sheets issued by the company provider of LAB starters, ST was added to the three starters for its thickening properties in plant-based liquid beverages [20]. The choice of the other two microbial starters was made according to their synergy with ST and the metabolic traits influencing the chemical and sensory features of the final products, more specifically, high acidification and potential to decrease the anti-nutritional factors, as indicated by the manufacturer company (SACCO srl, Cadorago, Italy), and observed in different legume-derived matrices [8,12,20,21,22,23]. All aliquots were incubated at 30 °C for 16 h and then stored in the same conditions at 4 °C. Analyses were conducted after 24 h of refrigeration. All samples had a pH of 6.10 before incubation. Three independent productions were carried out and monitored. The samples were analyzed in triplicate. 

### 2.3. Proximate Analyses 

The following methods were used: the AACC Soxhlet method 30-25.01 for the lipid content [24], EEC 2568/91 for the saturated fat content [25], the AACC method 08-0.1.01 for the ash content [26], as well as drying samples at 105 °C with a thermal balance MAC 110/NP RADWAG (Radom, Poland) for the moisture content, high-performance liquid chromatography coupled with a refractive index detector (HPLC-IR) for the sugar contents, (fructose, galactose, glucose and saccharose) as described by [27], and the AACC Kjeldahl method 46-13.01 for the protein content (N × 6.25) [28]. The results of the moisture, fat, protein and ash obtained were used to calculate the content of the carbohydrates content as follows:Carbohydrates = 100 − moisture − fat − protein − ash

The Atwater conversion factors were used to calculate the total energy and energy from proteins in kilocalories (kcal) [29].

Volatile organic compounds (VOC) were extracted by the solid-phase microextraction (SPME) technique. The analytes were separated and identified from all samples by a GC-MS system (TRACE 1300 gas chromatograph, ISQ Series 3.2 SP1 single-quadrupolemass spectrometer, Thermo Scientific, Rodano, Italy) equipped with a VF-WAX MS column (60 m × 0.25 mm, 0.25 m). Vials containing 1 g of each sample (added with an internal standard, 2-octanol purchased from Sigma Aldrich) were loaded into an autosampler Triplus RSH (ThermoFisher Scientific, Rodano, Italy) and each of them was incubated at 50 °C for 40 min. Then, a divinylbenzene/carboxen/polydimethylsiloxane (DVB/CAR/PDMS) 50/30 mm SPME fiber assembly (Supelco, Bellefonte, PA, USA) was inserted into the vial headspace for the extraction phase (60 min). The fiber was desorbed at 220 °C × 2 min with ultrahigh-purity helium (carrier gas operating at 1 mL/min flow rate) placed into the injection port in a splitless mode. The analysis conditions were: an initial oven temperature of 40 °C (maintained for 5 min) was ramped at 4 °C/min to 140 °C, then with a 10 °C/min increase rate to 210 °C and held at that temperature for 7.5 min for a total run time of 45 min. The mass detector was set as follows: detector voltage, 1700 V; source temperature, 250 °C; ionization energy, 70 eV; scan range, 33–200 amu. VOCs were tentatively identified by a comparison between the analyte mass spectra and reference mass spectra of the NIST library (NIST 2.0 mass-spectrometry database). A semi-quantification was done using the internal standard method, and the results were expressed as µg/L. 

### 2.4. Mineral Analysis

The contents of zinc (Zn), copper (Cu), potassium (K) and phosphorus (P) were determined as follows: samples were digested by a Multiwave 3000 (Anton Parr, Austria) based on the UNI EN 13805:2014 standards and analyzed by a NexION ICP Mass Spectrometer (Perkin Elmer, Waltham, MA, USA). Samples were mineralized with nitric acid (Suprapur, MERCK, Kenilworth, NJ, USA) in the amount of 10 mL/0.5 g sample. The process was carried out in Teflon containers at a temperature of 200 °C for 40 min by means of a high-pressure microwave method (MarsXPres, CEM, Stallings, NC, USA). Potassium was quantified by adding Cesium chloride/aluminium nitrate buffer solution acc. to Schuhknecht and Schinkel, in the respective concentrations of 50 g/L. Potassium was determined at the wavelength of 766.5. The mineral content is given in g kg^−1^ of the final product (d.m.). 

### 2.5. Total Titratable Acidity and Phytic Acids 

Total titratable acidity (TTA) was measured according to the AACC 02-31.01 method [30]. The TTA % was calculated by converting the volume (V) of NaOH using the following equation: TTA% = 10 × V NaOH × 0.009 × 0.1 sample weight × 100

Meanwhile, the phytic acids were determined using the Megazyme kit K-PHYT 05/07, following the manufacturer’s instructions.

### 2.6. PH, Color and Texture Analysis

The pH was measured with a pH meter edge^®^ HI2002 HANNA instrument (Colombus, OH, USA). Instrumental color evaluations of lightness (L*), green-red (a*) and blue-yellow (b*) were measured using a CM-600d Konica Minolta (Chiyoda, Tokyo, Japan) colorimeter. ΔE was calculated as follows:∆E = [(∆L*)2 + (∆a*)2 + (∆b*)2]1/2.

The back extrusion (BET) test was performed using a Z1.0 TN texture analyzer (Zwick Roell, Ulm, Germany) equipped with a 5 kg load cell and an extrusion disc (Ø = 40 mm) [31]. The sample (80 mL) was loaded into a cylinder of 50 mm in diameter and compressed at 1.0 mm s^−1^ to a depth of 50% of the product’s height. The reported values represent the averages of six replicates. Data were acquired by means of the TestXPertII version 3.41 software (Zwick Roell, Ulm, Germany). The parameters measured were firmness, which is the maximum positive force in compression; consistency, which is the positive area of the curve; cohesiveness, which is the maximum negative force of the curve; the viscosity index, which is the negative area of the curve.

### 2.7. Quantitative Descriptive Analysis (QDA) of Sensory Characteristics

The QDA of the sensory features involved the presentation of the four samples (CTRL, ST, STLL, STLP). An amount of 50 mL of each sample at 4 °C was randomly coded to a panel of thirteen members (7 males; 6 females, age range 25 to 52) who were recruited among students and staff of the Food Science and Technology unit of the University of Bari, Italy, based on their previous experience in the sensory evaluation as described by [32]. The study protocol followed the ethical guidelines of the laboratory. The pre-test sessions were carried out by professors who are also members of the Italian National Organization of Experts panel for cheese sensory (ONAF) to define the list of descriptors (Appendix A
Table A1). The panelists evaluated the appearance, orthonasal and retronasal sensations, mouthfeel, taste and texture on a 10-point hedonic scale where 0 represented the absence of the attribute, 1 represented the low intensity, and 9 represented the high intensity.

### 2.8. Statistical Analysis 

All data were subjected to statistical analyses; one-way ANOVA followed by Tukey’s HSD (honestly significant difference) test at α = 0.05 using the R software [33]. The formulations were replicated three times by producing three different slurries. Each sample was analyzed twice except for the texture parameters, which were repeated six times. Data were expressed as means ± SD.

## 3. Results

### 3.1. Biochemical Characterization of the Beverages

The effect of lactic fermentation on the moisture content, titratable acidity and macronutrient composition is shown in Table 1.

Before fermentation, the black chickpeas slurry had a pH of 6.1 and decreased up to 4.5 after incubation at 30 °C for 16 h with the three different starter mixtures, confirming the effectiveness of the fermentation process. The pH of the control group decreased up to 5.7, most probably due to spontaneous fermentation. As shown in Table 1, the moisture content was similar in all samples. The TTA values were significantly higher in the samples inoculated with different LAB cultures compared with the control. Among the fermented beverages, STLP had the highest value, followed by ST and STLL (Table 1).

Fermentation with LAB significantly decreased the content of sucrose in the fermented beverages compared with the control group, with STLP having the lowest value. ST had the highest fructose than the control and other fermented beverages, while the glucose content did not differ among all samples. 

LAB fermentation did not affect either the protein or the fat contents and the calculated calories from proteins; however, it lowered the saturated fat content (1.80–2.20 g kg^−1^ vs. 3.20 g kg^−1^). The calculated carbohydrate content was different between the formulated beverages, with ST having the highest value and the STLL and STLP having the lowest values.

After LAB fermentation, an increase in the contents of ash was observed from 12.2 in the CTRL to 13.2 g kg^−1^ in ST and STLP (Table 2). The minerals included in the study were those present in a considerable amount based on the recommended dietary intake (RDA). The starter mixtures affected the mineral contents differently. The fermentation with STLL and ST increased the content of K, and the fermentation of ST increased the content of Zn, while the fermentation with STLP lowered the content of Cu and K. 

Compared with the CTRL (639 g kg^−1^), the fermented beverages had a remarkable increase in P content (888–1230 mg kg^−1^), with ST having the lowest one and STLP having the highest. On the other hand, fermentation caused a reduction in the phytate content in all fermented beverages (1.40–6.09 vs. 6.68 g kg^−1^). The reduction was more evident in ST and STPL than in STLL, highlighting the different effects of the microbial strain in the phytate reduction.

### 3.2. Texture and Color

The differences in the texture and color between the formulated beverages are reported in Table 3.

After fermentation, all the texture parameters and lightness (l*) and redness indexes (a*) increased, while the yellow index (b*) decreased. 

In particular, the firmness did not differ between fermented beverages, while STLP was characterized by a considerably lower consistency, cohesivity and viscosity when compared with ST and STLL. Regarding the color index, STLP had the highest l* and the lowest a* among fermented beverages, while STLL had the highest b*. The ∆E of the LAB-fermented samples was calculated and compared to the CTRL in order to better evaluate the color differences between the samples. The ∆E was higher in STLP (4.05) than in ST (3.55) and STLL (3.41).

### 3.3. Sensory Analysis 

The spider diagram of the 16 sensory attributes for the four beverages is shown in Figure 2. Fermentation changed the taste/aftertaste and texture of the chickpea-based beverage considerably. In particular, the CTRL sample had the highest scores of grass and legumes aftertaste. In conformity with the fact sheet of *Lactococcus Lactis*, STLL was characterized by having a higher acid aroma (6.30) and astringent taste (6.00) and a lower sweet taste. As expected, LAB-fermented beverages had a higher sour taste score than the control, with STLL and STLP having higher scores than ST. The yogurt and yeast aftertastes were barely noted by the panelists, while the salt and better tastes were very mild in all samples.

In contrast with the TPA results, STLP obtained the highest score of creaminess (7.00). STLP, together with ST, had the highest score of stickiness (6.14 and 6.17). CTRL and STLL had a higher score for solubility than ST and STLP. LAB fermentation did not affect the attributes of homogeneity (mean score of 5.95) and adherence (mean score of 8.16). The CTRL sample had the most intense color (6.33), which ranged from beige to brown, while STLP showed the lowest one.

### 3.4. Identification of Volatile Compounds in Fermented Black Chickpeas Beverages

The analysis of the volatile compounds allowed the identification of numerous compounds (Table 4), many of which were recognized as off-flavor compounds according to the literature. 2-heptanone, octanal and octanoic acid were found in higher amounts in STLP. The former was absent in the CTRL, while its amount was similar in ST and STLL; octanal was found only in STLP, whereas octanoic acid was found in a higher amount in both STLP and STLL. 

STLL had the highest amount of heptanal, nonanal, 2,4-nonadienal and decanoic acid. Nonanal and decanoic acid were absent in the CTRL, and 2,4-nonadienal was absent in all the other beverages. 

1-penten-3-ol and dimethyltrisulfide were only found in ST. 2-pentylfuran was mainly found in ST, followed by STLP and the control, while it was absent in STLL. 

1-hexanol was the most abundant molecule in the table and was the only molecule found at a higher concentration in the CTRL than in the other samples. The CTRL was also characterized by its lowest concentration of 1-octen-3-ol, hexanal, 2-octenal and decanal.

## 4. Discussion

### 4.1. Biochemical Analyses

The success of fermentation was evaluated by acidification (pH and titratable acidity) and carbohydrate metabolization. The results revealed a decrease in sucrose after fermentation by 35–68%. During fermentation, sucrose is cleaved into fructose and glucose phosphate. ST showed the highest TTA and fructose concentration. This indicates that the fructose accumulated was subsequently less consumed by ST when compared with STLL and STLP. In fact, *Streptococcus thermophilus* was shown to preferably metabolize lactose, whereas the uptake of other sugars, including sucrose, fructose and glucose, depends on the specific strains [34]. On the other hand, *Lactococcus lactis* has a strong fructokinase activity [35]. In contrast, the *Lactobacillus plantarum* strains showed higher survival, adaptation and fermentation abilities than any other lactic acid bacteria, confirming our TTA and sugars results [36].

The protein content of the formulated beverages, around 120 g kg^−1^, was higher than almost all PB beverages existing on the market and in literature, which represents a quality marker of the slurry [4,5,11,37]. According to the EU, a claim ‘rich in proteins’ can be given to all beverages. This highlights the importance of using dry-fractionated legumes flour for the production of food rich in protein [32,38]. A ‘low in fat’ claim could also be assigned to the formulated CP beverages, as they contained less than 3% fat [38]. The amount of fat, 1.7%, was higher than that of chickpeas-based beverages (0.74%) prepared by Wang et al. [39] and is considered a sufficient source for lipid oxidation by LOX and consequently an off-flavor formation [40], as observed in the VOC analysis. 

Adebo et al. [39] explained the mechanisms of nutrient modifications after legume fermentation and showed that for each nutrient, an increase and a decrease in contents could happen simultaneously. In our study, fermentation did not have a significant effect on the protein and fat contents. The increase in protein content that could be seen after the release of the protein chelated by anti-nutritional compounds, or the increase in fat content after an increase in lipase/lipolytic enzymes, could be counterbalanced by the use of carbohydrates and fat by the fermenting microorganisms as an energy source [41]. In this study, LAB fermentation decreased 30–43% the content of saturated fat, an indication of an increase in the polyunsaturated fatty acids content by fermenting LAB in agreement with the results obtained by Ziarno et al. [42]. 

It is still not clear whether fermentation decreases phytates by decreasing the pH and thus activating plant phytases (ideal pH 4–5) or by producing microbial phytases [43], or it could be by using both ways. In order to determine the effect of fermentation on phytic acid, its content was determined in the CTRL and fermented beverages. *Streptococcus thermophilus* in monoculture and in a co-culture with *L. plantarum* was able to considerably decrease phytic acid (PA) up to 69% and 79% of the initial content, while *S. thermophilus* in a co-culture with *Lactococcus lactis* decreased PA only by 9%. In fact, the sugar contents and titratable acidity analyses revealed that the presence of *S. thermophilus* in a co-culture with *L. plantarum* increased the acidification more than the other two starter mixtures (a consequence of more lactic acid production). This could explain, in part, the higher effect of STLP on phytic acid. Our results agree with those of Pontonio et al. [9], Rui et al. [44], Starzyńska-Janiszewska and Stodolak [45], Fritsch et al. [46], and Zamudio et al. [47], who reported that L. plantarum considerably decreases phytic acid in legumes, attributing the result to L. plantarum producing microbial phytases. On the other hand, our results contrast with those of dry-fractionated fava beans flour fermented by L. plantarum by Coda et al. [48]. This discrepancy could be attributed to the important synergy between the starter culture species and the type of substrate in the fermentation performance. Moreover, our results on STLL could be in agreement with Sreeramulu et al. [49], who showed that *Lactococcus lactis* does not produce microbial phytases. Lai et al. [50] reported that *Streptococcus thermophilus,* in co-fermentation, is able to degrade phytates in legumes. We could add that even in monoculture, *Streptococcus thermophilus* considerably decreases the phytase present in black chickpeas slurry treated at 90 °C and fermented for 16 h at 30 °C, reaching a pH of 4.5. 

Phytate represents the major storage form (phytate P) of phosphorus (P), and thus, any increase in phytate degradation is associated with an increase in the P content. This explains the great increase in phosphorus content by 30% and 34% after the STT and STLL fermentations, respectively, and by 48% after the STLP fermentation. However, no significant correlation was found between the phytic acid and phosphorus contents in the samples studied, and this could be due to phytate P representing 50% to 85% of the total P in legumes and the presence of other anti-nutrient compounds binding P, such as oxalates [51].

As for phosphorus, LAB fermentation produces organic acids that could bind to the minerals chelated with phytates and, thus, make them available to phytases, confirming the increase seen in the Cu, Zn and K contents after fermentation [52,53,54]. According to the 2015–2020 dietary guidelines, potassium is considered a ‘nutrient of public health concern’, and its adequate intake is of great importance. The potassium content of our samples fits in the list of foods high in potassium, according to the USDA, and so could be considered a source of potassium [55]. A product is considered a micronutrient source if 15% of the daily reference intake is supplied by 100 g of the product [56]. Accordingly, besides K, the prepared samples in this study were a good source of Cu and Zn, whereas only STLP was a good source of P [57]. 

### 4.2. Texture and Color Analyses

One of the main challenges faced today by the PB food industry is to achieve a pleasant visual appearance and texture attributes of the end-products. 

Considerable changes in the overall texture were observed after LAB fermentation, as suggested by SACCO and in accordance with Li et al. [58] and Xu et al. [59]. LAB starters increased firmness by 37% to 42%. Behare et al. [60] studied the technological performance of 64 EPS-producing LABs isolated from Lactobacillus, Streptococcus and Enterococcus genera and identified a strain from *Streptococcus thermophilus* to be promising in improving the viscosity, flavor, consistency, color and appearance of fat-free yogurt drinks. In our study, the ST and STLL fermentations increased the consistency, cohesivity and viscosity of the beverages more than the STLP fermentation. Li et al. [58] attributed the difference in textural improvement after different LAB fermentations to the difference in the exopolysaccharide (EPS) amounts produced. In particular, the network created by EPS and protein could be, in part, responsible for the texture improvement, especially for the viscosity. However, a reduction in viscosity after reaching its maximum was reported by Mengi et al. [61], and the decrease was attributed to a degradation of the hetero-exopolysaccharide produced, thus resulting in a partial loss of the network. The higher acidification produced by STLP could be behind the degradation of the EPS produced and, hence, the decrease in consistency and viscosity observed. The EPS production by the dairy LAB varies from strain to strain and is affected by several factors, among them the composition of the medium. Using black chickpeas as a medium and under the conditions of our study (pH of 4.5, T 30 °C), ST in monoculture and in a co-culture with *Lactococcus lactis* showed to enhance cohesivity, consistency and viscosity more than in a co-culture with *Lactobacillus plantarum*. 

To understand better if the changes in texture observed were favorable or not, we have compared our results with the one obtained by Grasso et al. [62], who analyzed the textural parameters of soy, coconut, cashew and almond yogurt analogs and compared them with commercial dairy yogurt. Compared with the results of Grasso et al. [62], the firmness and cohesiveness of ST, STLL and STLP were higher. The consistency of the STLP beverage was similar to the dairy yogurt, while the consistency of the ST and STL beverages was in the range of those of the PB yogurt analogs. The viscosities of ST, STLL and STLP were in the range of those of the PB yogurt analogs. 

Fermentation increased the lightness of the three fermented beverages in contrast with findings by Shi et al. [63]. The ∆E values were between 3.5 and 4.05, meaning that the difference between samples is clearly detected [64]. 

### 4.3. Sensorial and VOC Analysis

The resulting acidification of food by LAB fermentation improved the sensory profile, nutritional properties, texture and microbial safety of PB food, avoiding the addition of chemical preservatives (clean label) while preserving their original natural status [8,10]. In our study, LAB fermentation considerably lowered the beany and grass aromas, even though it did not decrease the level of unpleasant VOCs. This could be due to LAB fermentation generating new compounds that masked, in particular, the beany and grass aromas, such as the sour taste, which is in accordance with [8,58,65]. The variation in the organoleptic properties was considerable for the three LAB-fermented beverages investigated. Even though STLP fermentation was responsible for the highest acidity content, the panelists rated the sourness of STLL and STLP as equal and the acidity aroma to be the highest in STLL. Regarding the texture attributes, and in accordance with the back extrusion texture results, STLP fermentation increased the lightness of color and the creaminess of texture more. Both ST as a monoculture and in a co-culture with LP increased the stickiness more and decreased the solubility of the final product more. 

Usually, LAB fermentation is used as a strategy to mask or eliminate the undesired flavor of legume-derived ingredients and food products [8]. Nevertheless, few examples of unpleasant flavor formation after LAB fermentation were highlighted [65]. Undoubtedly, the contribution of LAB to the volatile profile was found by increasing the amount of the already existing compounds and generating new ones. Since samples were subject to the same thermal treatment, the absence of 2-heptanone in the control and its presence in fermented samples could be attributed to the fermentation process by the specific strains chosen. Pulses and Apulian black chickpeas, in particular [16], are rich in unsaturated fatty acids (e.g., linoleic and linolenic acid), as well as lipid-modifying enzymes, and thus, many volatile short-chain aldehydes, ketones, hydrocarbons and alcohols responsible for the off-flavor could be produced [66,67,68,69]. Aldehydes are commonly related to the legumes’ off-flavor and are produced by fatty acid oxidation [68]. According to the literature, hexanal is typically found in peas conferring fatty, green and grassy odors, and the higher its quantity, the less the acceptability of the product by consumers [70]. In contrast with Achouri et al. [71], in our study, neither Streptococcus alone nor in a co-culture with Lactobacillus plantarum or Lactococcus lactis eliminated the hexanal and 2-pentylfuran, which on the contrary, they increased in the LAB-fermented samples, and mostly in ST. Despite 2-pentylfuran contributing to the off-flavor in soy products, it could also suppress the perception of other off-flavor molecules, such as dimethyltrisulfide, usually ascribed to the odor of feces, meat broth and sewers, in peas [68,72]. Since 2-pentylfuran was absent in STLL and found in the control, its formation is not strictly attributed to LAB fermentation but by other routes, such as those reported by Bradley et al. [73]. Acids, such as octanoic and decanoic acids, could arise from the auto-oxidation of linoleic acid and were attributed to a strong unpleasant odor in chickpeas, such as the sweaty flavor of octanoic acid [68,72]. In light of all this, it is worth saying that the increase or decrease trends for a single active aroma compound responsible for the off-flavors are important. However, the odor perception of foods is not only caused by a single VOC; in fact, it is difficult to link sensory perception to a single volatile in a mixture of different compounds [74]. Many studies focused on the odor perception given by a combination of different molecules [75]. For example, omitting 1-pentanol, known to contribute to beany off-flavors, from a mixture of molecules did not modify the beany perception of the mixture; on the other hand, the omission of hexanal or 2-octenal led to a different aroma perception [75]. In addition, the concentration of a single volatile molecule in a food product plays a pivotal role in odor perception. Finally, the results obtained in our study are in agreement with Schindler et al. [40], who highlighted the existing differences of the volatile profile and odor perception of LAB-fermented and control samples.

## 5. Conclusions

Many types of legumes, beans, flours and isolates were studied for their suitability in formulating PB-fermented beverages as an alternative to yogurt [76]. However, no studies are available on the use of legume protein concentrates obtained by dry-fractionation in PB-fermented beverages. 

The results of the study indicate that the Apulian black chickpeas could be a potential food carrier for lactic acid bacteria and that dry-fractionation exhibits the potential to produce a product high in protein and low in carbohydrate contents. The starter cultures selected were able to decrease the phytates and saturated fatty acids content and increase the lightness, consistency, cohesivity, viscosity and creaminess of the formulated products. Moreover, the results demonstrated that LAB fermentation, through an increase in lactic acid and exopolysaccharides productions and the inter- and intra-molecular interactions of macromolecules of starch and proteins, could increase the consistency and viscosity of liquid and semi-liquid products to a certain extent, after which these parameters might decrease. On a sensorial level, LAB fermentation did not decrease the main VOC responsible for undesired flavors; however, it created other molecules which could have balanced the complex mixture of flavors in the final products. 

Optimization of the flavor is still needed. However, taken together, these results demonstrated that the fermented black chickpeas slurry is a very promising innovative product to be considered on an industrial level as an alternative to yogurt or as a base matrix for further food development, such as plant-based cheese or ice cream.

## Figures and Tables

**Figure 1 foods-11-03578-f001:**
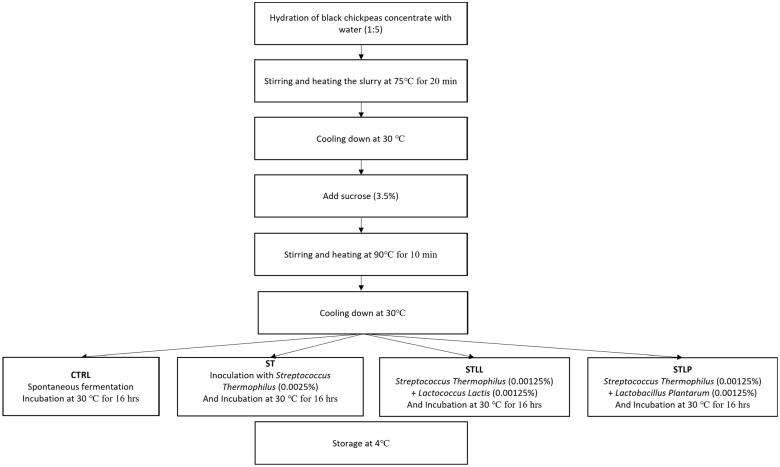
Protocol of preparation of the three black chickpeas fermented beverages.

**Figure 2 foods-11-03578-f002:**
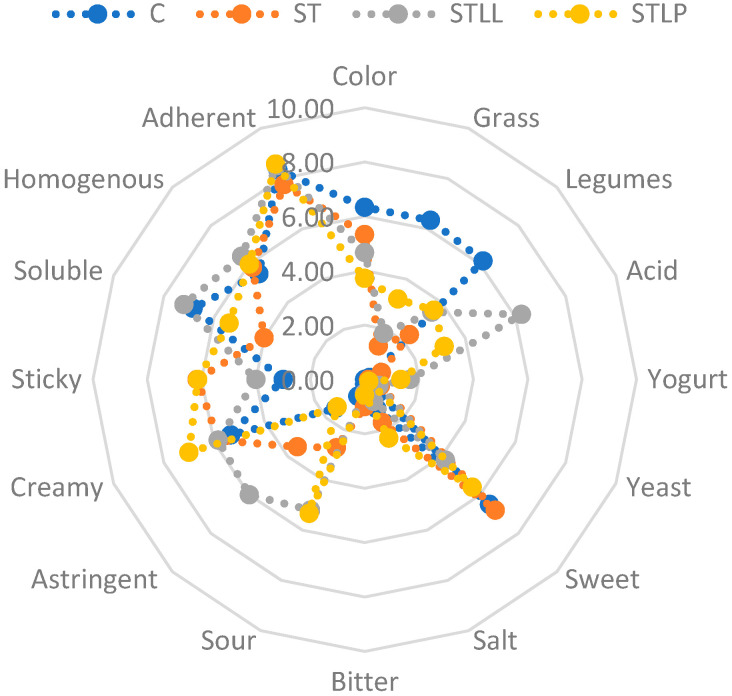
The spider diagram of descriptive sensory analysis on the beverages made with black chickpeas and fermented with Streptococcus thermophilus (ST), represented by orange dots and line, Streptococcus thermophilus and Lactococcus lactis (STLL), represented by grey dots and line and Streptococcus thermophilus and Lactobacillus plantarum (STLP), represented by yellow dots and line. A beverage without inoculum (CTRL) was used as control and is represented by the blue dots and line.

**Table 1 foods-11-03578-t001:** Total titratable acidity (TTA) (%), moisture content (g kg^−1^), sugars concentration (g kg^−1^), fat and saturated fat (g kg^−1^), proteins (g kg^−1^), calculated carbohydrates (g kg^−1^) and calories from proteins (kcal) in beverages made with black chickpeas and fermented with *Streptococcus thermophilus* (ST), *Streptococcus thermophilus* and *Lactococcus lactis* (STLL) and *Streptococcus thermophilus* and *Lactobacillus plantarum* (STLP). A beverage without inoculum (CTRL) was used as a control. Data are expressed as means ± SD.

	CTRL	ST	STLL	STLP
TTA	0.90 ± 0.70 ^d^	1.65 ± 0.70 ^b^	1.45 ± 0.00 ^c^	1.80 ± 1.40 ^a^
Moisture	726 ± 0.62 ^a^	721.5 ± 18.0 ^a^	731.5 ± 2.10 ^a^	733 ± 11.4 ^a^
Sucrose	30.2 ± 1.40 ^a^	19.7 ± 0.80 ^b^	11.1 ± 0.40 ^c^	9.60 ± 0.60 ^d^
Fructose	0.20 ± 0.00 ^b^	2.40 ± 0.00 ^a^	0.20 ± 0.00 ^b^	0.30 ± 0.00 ^b^
Glucose	0.10 ± 0.00 ^a^	0.20 ± 0.0 ^a^	0.10 ± 0.00 ^a^	0.20 ± 0.00 ^a^
Fat	17.2 ± 0.90 ^a^	16.8 ± 0.60 ^a^	16.7 ± 0.40 ^a^	17.8 ± 1.20 ^a^
Saturated fat	3.20 ± 0.20 ^a^	2.20 ± 0.30 ^b^	1.90 ± 0.10 ^b^	1.80 ± 0.40 ^b^
Carbohydrates	127 ± 0.40 ^b^	131 ± 1.40 ^a^	116 ± 0.40 ^c^	115 ± 0.40 ^c^
Proteins	118 ± 1.40 ^a^	117 ± 1.50 ^a^	121 ± 3.20 ^a^	122 ± 7.02 ^a^
Calories from proteins	47.9 ± 0.14 ^a^	48.64 ± 0.49 ^a^	47.6 ± 0.21 ^a^	47.35 ± 0.21 ^a^

Means with the same letter are not statistically different with Tukey’s test at *p* ≤ 0.05.

**Table 2 foods-11-03578-t002:** Ash (g kg^−1^), copper (Cu) (mg kg^−1^), potassium (K) (mg kg-1), zinc (Zn) (mg kg^−1^), phosphorus (P) (mg kg^−1^) and phytic acid (g kg^−1^) contents of beverages made with black chickpeas and fermented with *Streptococcus thermophilus* (ST), *Streptococcus thermophilus* and *Lactococcus lactis* (STLL) and *Streptococcus thermophilus* and *Lactobacillus plantarum* (STLP). A beverage without inoculum (CTRL) was used as a control. Data are expressed as means ± SD.

	CTRL	ST	STLL	STLP
Ash	12.2 ± 0.30 ^b^	13.2 ± 0.20 ^a^	12.5 ± 0.80 ^ab^	13.2 ± 0.30 ^a^
Cu	3.50 ± 0.00 ^a^	3.60 ± 0.00 ^a^	3.60 ± 0.00 ^a^	2.90 ± 0.00 ^b^
K	4720 ± 30 ^b^	4860 ± 100 ^a^	4930 ± 90.3 ^a^	3870 ± 70.0 ^c^
Zn	103 ± 2.80 ^b^	113 ± 5.10 ^a^	111 ± 7.00 ^ab^	102 ± 3.40 ^b^
P	639 ± 26.1 ^d^	888 ± 25.4 ^c^	972 ± 16.2 ^b^	1230 ± 40.1 ^a^
Phytic acid	6.68 ± 0.08 ^a^	2.08 ± 0.16 ^c^	6.09 ± 0.05 ^b^	1.40 ± 0.19 ^d^

Means with the same letter are not statistically different with Tukey’s test at *p* ≤ 0.05.

**Table 3 foods-11-03578-t003:** Textural parameters, firmness (N), consistency, cohesivity and viscosity (Pa s), and color indexes, lightness (L*), red/green (a*) and yellow/blue (b*) of beverages made with black chickpeas and fermented with *Streptococcus thermophilus* (ST), *Streptococcus thermophilus* and *Lactococcus lactis* (STLL) and *Streptococcus thermophilus* and *Lactobacillus plantarum* (STLP). A beverage without inoculum (CTRL) was used as a control. Data are expressed as means ± SD.

	CTRL	ST	STLL	STLP
Firmness	1.63 ± 0.06 ^b^	2.70 ± 0.04 ^a^	2.83 ± 0.03 ^a^	2.60 ± 0.10 ^a^
Consistency	2.79 ± 0.51 ^c^	13.0 ± 0.14 ^a^	11.3 ± 0.61 ^a^	5.47 ± 0.92 ^b^
Cohesivity	0.81 ± 0.05 ^c^	1.64 ± 0.03 ^a^	1.77 ± 0.06 ^a^	1.38 ± 0.06 ^b^
Viscosity	3.39 ± 0.26 ^c^	11.3 ± 0.02 ^a^	10.4 ± 0.65 ^a^	6.37 ± 0.15 ^b^
L*	68.50 ± 0.01 ^d^	71.4 ± 0.01 ^c^	71.8 ± 0.01 ^b^	72.5 ± 0.01 ^a^
a*	3.00 ± 0.01 ^d^	3.49 ± 0.02 ^b^	3.59 ± 0.01 ^a^	3.16 ± 0.02 ^c^
b*	26.6 ± 0.01 ^a^	24.7 ± 0.02 ^d^	26.1 ± 0.02 ^b^	25.7 ± 0.02 ^c^
∆E vs. CTRL		3.55 ± 0.02	3.41 ± 0.02	4.05 ± 0.02

Means with the same letter are not statistically different with Tukey’s test at *p* ≤ 0.05.

**Table 4 foods-11-03578-t004:** The volatile organic compounds responsible for undesired flavors of beverages made with black chickpeas and fermented with *Streptococcus thermophilus* (ST), *Streptococcus thermophilus* and *Lactococcus lactis* (STLL) and *Streptococcus thermophilus* and *Lactobacillus plantarum* (STLP). A beverage without inoculum (CTRL) was used as a control. Data are expressed as means ± SD.

	STLP	STLL	ST	CTRL
2-eHptanone	2.77 ± 0.23 ^a^	1.04 ± 0.10 ^b^	0.89 ± 0.14 ^b^	0.00 ± 0.00 ^c^
1-Penten-3-ol	0.00 ± 0.00 ^b^	0.00 ± 0.00 ^b^	0.64 ± 0.04 ^a^	0.00 ± 0.00 ^b^
1-Pentanol	16.4 ± 0.60 ^a^	0.00 ± 0.00 ^b^	17.8 ± 1.80 ^a^	16.14 ± 0.6 ^a^
2-Penten-1-ol	1.11 ± 0.28 ^a^	1.76 ± 0.50 ^a^	0.95 ± 0.31 ^a^	0.97 ± 0.36 ^a^
1-Hexanol	121 ± 8.10 ^c^	108 ± 7.3 ^c^	155 ± 4.83 ^b^	166 ± 6.10 ^a^
1-Octen-3-ol	36.5 ± 3.44 ^a^	29.9 ± 4.81 ^a^	30.48 ± 2.78 ^a^	11.5 ± 1.40 ^b^
Hexanal	59.8 ± 3.92 ^b^	71.9 ± 9.6 ^ab^	95.7 ± 18.3 ^a^	20.9 ± 4.29 ^c^
Heptanal	4.01 ± 0.92 ^b^	14.8 ± 0.32 ^a^	2.14 ± 0.99 ^bc^	1.39 ± 0.20 ^c^
Octanal	5.2^2^ ± 0.07 ^a^	0.00 ± 0.00 ^b^	0.00 ± 0.00 ^b^	0.00 ± 0.00 ^b^
Nonanal	34.^5^ ± 5.17 ^b^	68.1 ± 12.8 ^a^	14.1 ± 4.37 ^c^	0.00 ± 0.00 ^d^
2-Octenal	12.8 ± 2.94 ^ab^	8.90 ± 0.98 ^b^	15.1 ± 3.45 ^a^	3.96 ± 0.58 ^c^
Decanal	3.17 ± 0.85 ^a^	3.27 ^±^ 0.87 ^a^	1.97 ± 0.46 ^a^	0.00 ± 0.00 ^b^
2,4-Nonadienal	0.00 ± 0.00 ^b^	1.18 ± 0.20 ^a^	0.00 ± 0.00 ^b^	0.00 ± 0.00 ^b^
2-Pentylfuran	7.81 ± 1.10 ^b^	0.00 ^±^ 0.00 ^d^	11.04 ± 1.12 ^a^	2.79 ± 0.49 ^c^
Octanoic acid	7.16 ± 1.28 ^a^	3.60 ± 0.47 ^b^	1.57 ± 0.44 ^c^	1.64 ± 0.50 ^c^
Decanoic acid	3.52 ± 0.41 ^b^	11.2 ± 0.87 ^a^	0.21 ± 0.05 ^c^	0.00 ± 0.00 ^d^
Dimetyl trisulfide	0.00 ± 0.00 ^b^	0.00 ± 0.00 ^b^	1.10 ± 0.14 ^a^	0.00 ± 0.00 ^b^

Means with the same letter are not statistically different with Tukey’s test at *p* ≤ 0.05.

## Data Availability

Data is contained within the article.

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
