# Peer review of "Development and Chemical-Sensory Characterization of Chickpeas-Based Beverages Fermented with Selected Starters"

_foods, 2022, doi:10.3390/foods11223578_

Round 1

Reviewer 1 Report

The paper contains new and interesting information and it is carefully written and well organized. The experimental design looks robust and the conclusions are supported by the data presented

A few observations:

The pH of the black chickpeas slurry ( pH of 6.1) decreased up to 4.5 after incubation at 30 °  for 16 hours with the three different starter mixtures. How do you explain the pH decrease of the control group to 5.7?

Rewrite the last part of the conclusions making it more clear and objetctive: “should discuss the results and how they can be interpreted from the perspective of previ-496ous studies and of the working hypotheses. The findings and their implications should be discussed in the broadest context possible. Future research directions may also be high-lighted.”

Some sentences have some unnecessary commas. Please revise

Author Response

Please attachment.

Reviewer 2 Report

The authors of manuscript FOODS-1973305 evaluated the development and chemical-sensory characterization of chickpeas-based beverage fermented with selected starters. This work provided an opportunity to develop a clean label and safe plant-based fermented beverage, but the conclusions of this draft could not be supported by favorable evidence, and the writing of the manuscript made it impossible to deeply understand the background of the study and analyze the results in depth.

First of all, the three “biotechnology” is too broad, and the work uses only LAB fermentation of three different starter cultures, suggesting a different way of describing it.

Then, section 3.1 states that the ST group had higher fructose content than the control group, but the discussion section does not explain the causes of this phenomenon.

Finally, it was concluded that LAB fermentation reduced the content of phytate and saturated fatty acids and increased the phosphorus content, and the effect of STLP fermentation was more obvious, which was played by Streptococcus thermophilus or Lactobacillus plantarum. Why not do LAB fermentation with only Lactococcus lactis and Lactobacillus plantarum single starter cultures.

Specific comments:

Line 13 : “Streptococcus thermophilus”, “Lactococcus lactis” and “Lactobacillus plantarum” should be italicized. Specific name should be italicized. The same mistakes happen in line 14, 57, 76, 87, 99, 100, 221, 242, 243, 270, 271, 311, 312, 363, 364, 392.

Round 2

Reviewer 2 Report

The manuscript has been well revised and I have no further questions